**communications** engineering

# Assessment of environmental impacts and costs for hydrogen-powered cargo shipping in Europe
Simon Schlehuber [1]✉, Moritz Gutsch [1], Niklas Kronemeyer [1,2], Florian Frieden [1,3] & Stephan von Delft [4,5]

The transportation of goods contributes considerably to global greenhouse gas emissions, underscoring the urgency of sustainable transport solutions. A promising yet underexplored solution is the introduction of small autonomous-driving hydrogen-powered boats (AHB) that can substitute long-haul trucking. Here we address this gap by modelling an AHB and then performing a life cycle assessment, coupled with a total cost of ownership analysis, across various scenarios. AHBs powered by green hydrogen are expected to emit 0.46 kg $CO_2$ eq $km^{-1}$ and cost 0.82 € $km^{-1}$ on average. In contrast, AHBs powered by gray hydrogen are characterized by average emissions of 1.12 kg $CO_2$ eq $km^{-1}$ and costs of 0.42 € $km^{-1}$. Furthermore, costs of the modelled AHB are compared to those of semi-trucks powered by different fuels to gauge the AHB's real world applicability. The results show that an AHB could be the cost-optimal solution for non-time sensitive transportation of goods exceeding distances of 624 km.

Globally, the premature death of 4.2 million individuals has been attributed to air pollution, a phenomenon inextricably linked to the activities of the transport sector[1]. Furthermore, this sector contributes significantly to anthropogenic climate change, accounting for 14% of total global greenhouse gas emissions between 2010 and 2018[2]. Given that global freight transportation is at an all-time high[3,4] and is expected to continue to grow by 5.6% annually until 2028[3], establishing and implementing net-zero carbon dioxide ($CO_2$) strategies in the transport sector is more crucial than ever.

In light of these pressing concerns, recent research has explored the feasibility of long-distance freight shipping propelled by alternative propulsion systems such as hydrogen fuel cells, batteries, and biofuels[5–7]. Similarly, there is a discernible emphasis on alternative propulsion systems for short to medium-distance transportation ranging from 0 to 805 km (0 to 500 miles), a domain commonly serviced by road trucking[8]. The electrification of semi-trucks, either through batteries or fuel-cells with hydrogen storage, is starting to take off[9–12]. Given the complexity of the global transport sector, a single solution seems unlikely[13]. Instead, multiple technologies will be necessary for a smooth transition to a more environmentally friendly transport sector.

To contribute to the field of transport research, this study focuses on a potential alternative for long-haul trucking: the emerging technology of autonomous-driving hydrogen-powered boats (AHBs). These boats could compete with semi-trucks for the transportation of forty-foot equivalent unit (FEU) containers (one of the most common means for transporting goods)[14–16]. While AHBs, like other clean technologies in the early stages of their evolution, face challenges such as technological and legal uncertainty as well as a lack of (hydrogen refueling) infrastructure, they have potential for reducing carbon emissions in short to medium-distance freight transport. AHBs may also offer superior total cost of ownership, a characteristic that is positively influenced by its autonomous operation[12].

Furthermore, compared to larger river freight ships, which are typically about 135 m long, 15 m wide, and 3 m deep[17], the depth of the river required for AHB operation is around 50% lower, thereby expanding accessibility to river system in countries such as Germany, the Netherlands, and the United States. Moreover, the decreased draft of AHBs would prove advantageous in periods of drought, which are anticipated to become more common because of climate change[18]. Under such conditions, reduced river depths may make safe navigation of larger river barges impossible[18]. AHBs could therefore help stabilize river trade during droughts and provide an alternative to road transport. To put a possible expansion of the accessible river network into perspective, Fig. 1 provides an overview of an exemplary river system in Europe[19].

[1]Institute of Business Administration at the Department of Chemistry and Pharmacy, University of Münster, Leonardo Campus 1, Münster, Germany. [2]Porsche Consulting GmbH, Siemensstraße 6, Stuttgart, Germany. [3]FutureCamp Climate GmbH, Aschauer Straße 30, München, Germany. [4]University of Münster, REACH EUREGIO Start-up Center, Geiststrasse 24, Münster, Germany. [5]University of Glasgow, Adam Smith Business School, Glasgow, UK. ✉e-mail: simon.schlehuber@uni-muenster.de

**Fig. 1 | Overview of the river network in 29 European countries.** The color and width of the river corresponds to its scale rank (rather subjective factor that describes the relative relevance of the river based on characteristics like size, cultural importance, or political importance). Rivers shown in black are the most important for shipping, rivers in blue may still be relevant for shipping, while rivers in red are the least important or not used at all for current commercial shipping because of their limited size. Rivers with scale ranks above 12 were disregarded. This illustration is based on publicly accessible data from Natural Earth[19] (2024).

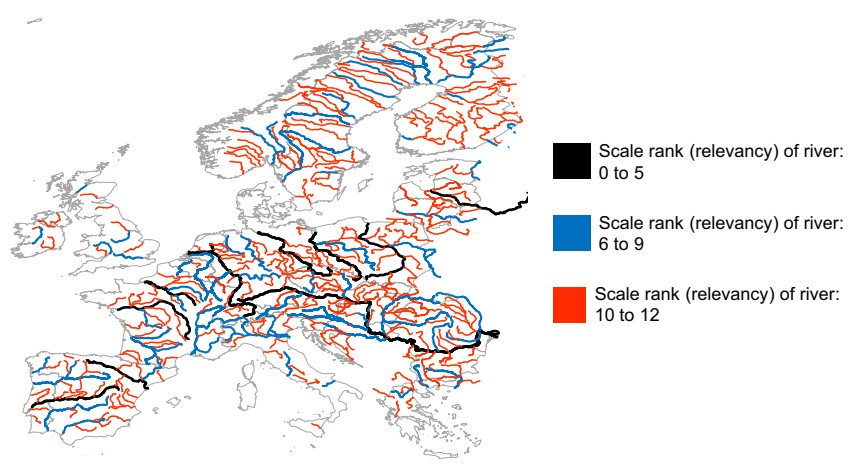

**Fig. 2 | Autonomous-driving hydrogen-powered boat (AHB) design. a** Technical model of the AHB without auxiliaries. The catamaran is made up of a 1.5 cm thick deck and two 20 m long hulls which consist of two triagonal prisms. **b** Illustration for a catamaran design. Image reprinted with permission from Unleash Future Boats GmbH.

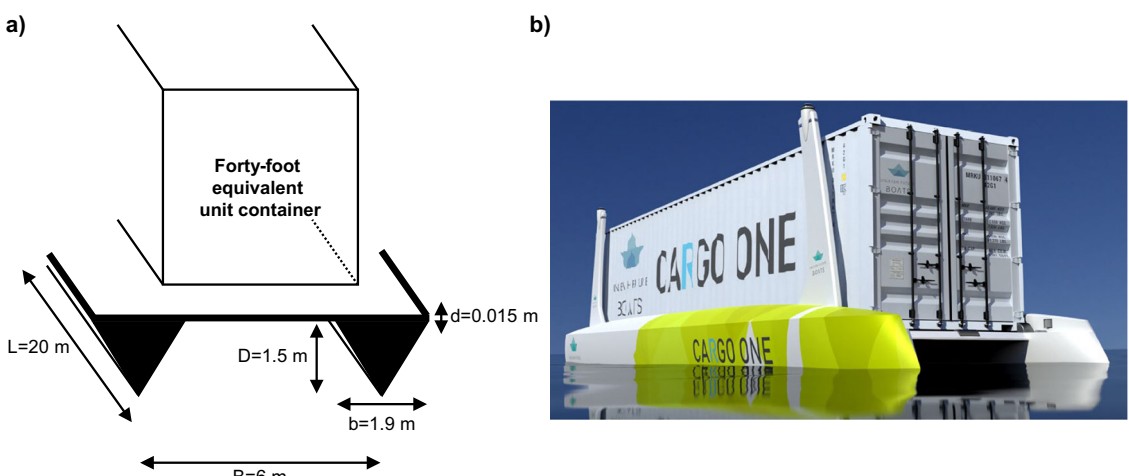

Although AHBs have the potential to become an important piece in the puzzle to reduce greenhouse gas emissions in the transport sector, an assessment of their environmental impacts and economic viability is thus far missing. Consequently, this study offers two main contributions. First, it models an AHB to provide a comprehensive picture of the environmental impacts and costs. By evaluating several design options with different transport speeds and power requirements for the propulsion system, this study provides strategic insights into the development of AHBs that effectively minimize both environmental impacts and costs. Second, by benchmarking the AHB against reference semi-trucks powered by diesel fuel, batteries, and hydrogen fuel cells, this study is the first to derive potential applications of AHBs for the decarbonization of the road freight sector.

## Results and Discussion
### Model of the AHB
Following recent developments in industry[15,16], this study models the AHB as a catamaran. Benefits of catamarans are a shallower draft, greater stability, and higher average speed due to having less water drag. However, catamarans may exhibit reduced maneuverability and higher initial costs in comparison to monohulls[20]. Overall, the lower water drag was the decisive criterion for choosing a catamaran over a monohull in this study.

The model of the catamaran is shown in Fig. 2. The AHB is 20 m in length, 6 m in width and 1.5 m in depth. Its dimensions greatly influence the wetted surface area (area of the boat that is in direct contact with water) and the total drag coefficient (coefficient that measures drag induced on boat by

water), which deserve special attention, because of their importance in the total cost of ownership (TCO) and life cycle assessment (LCA) calculations. The AHB and underlying calculations are presented in more detail in supplementary note 1 of the supplementary materials.

### Total cost of ownership analysis
To carry out a TCO analysis, the relevant cost parameters need to be defined. In the category of capital expenditures this includes the aluminum hull, the proton-exchange membrane fuel cells powered by hydrogen, the electric motor, and the hydrogen storage. Operational expenditures are broken down into fuel costs and loading costs. Here, fuel costs depend on the type of hydrogen: the cheaper gray hydrogen and the more expensive green hydrogen. Gray hydrogen is produced from natural gas or methane via steam reforming and the resulting greenhouse gases are not captured[21]. Green hydrogen is usually generated from water via electrolysis using renewable energy[21]. The higher the ratio of renewable to non-renewable energy, the lower the carbon footprint of the green hydrogen[22]. Loading costs are defined as a fixed monetary value and describe the costs of loading and unloading a container from the AHB per trip. They, therefore, capture costs for personnel involved and the use of port facilities. This study assumes fully autonomous operation of the boat, which is why there are no staff costs for navigating the boat. Recycling costs reflect the costs for the disposal of end-of-life material which is assumed to be entirely aluminum scrap to reduce complexity. As a final cost parameter, maintenance costs based on total capital expenditures are considered. The sum of all cost parameters is

then normalized by utilizing the kilometers of AHB travel per year, thus making up the levelized cost of ownership per km.

## Life cycle assessment

To compare the LCA and TCO analyses, the results of the LCA are normalized on a per-kilometer-per-year basis, analogous to the TCO analysis. Accordingly, the fixed environmental impact measured in global warming potential 100 (GWP100) is calculated by summing up the individual GWP100 of the AHB's hull, the hydrogen fuel cells, the electric motor and the hydrogen storage (see methods section and Tables S2-S4 in supplementary note 2 for more information on all impact categories). The variable environmental impact can then be determined by looking at the GWP100 of the fuel (gray vs. green hydrogen) as well as the process of loading. Gray hydrogen has a much larger GWP100 than green hydrogen owing to its route of synthesis. The GWP100 for recycling is calculated by modelling scrap aluminum incineration. After including a maintenance factor for the fixed environmental impacts, the GWP100 per km of AHB travel is calculated.

## Scenarios for the combined results of TCO analysis and LCA

This study considers four different scenarios which are based on four discrete boat speeds while keeping all other parameters constant (see supplementary note 2 in the supplementary materials). The speeds considered in the four scenarios are 5 km h$^{-1}$, 10 km h$^{-1}$, 15 km h$^{-1}$, and 20 km h$^{-1}$ and describe the speed of the AHB as seen on the global positioning system (GPS). This means that a boat must go faster against the river current and slower while travelling with the river current to maintain a specific GPS speed, therefore altering its maximum necessary power output. These speeds were chosen as they closely correspond to the average velocities (10 to 15 km h$^{-1}$) of common inland waterway vessels[23–25]. Important parameters include a trip distance of 805 km (500 miles) and a cost of 7 € kg$^{-1}$ for green hydrogen so that the data can be compared to the literature values of different types of trucks[26] as well as a gray hydrogen cost of 2 € kg$^{-1}$ [27,28]. The results of the combined TCO and LCA analysis are shown in Fig. 3. Generally, it can be inferred that costs and GWP100 rapidly increase when the speed of the AHB increases. This is due to several factors, such as strongly increasing power requirements, thus leading to additional costs for the fuel cell stack, electric motor, and hydrogen storage, greatly increased fuel usage and more frequent stops because trips are finished faster. The difference between green and gray hydrogen is also evident. While the TCO model indicates a significant advantage for gray hydrogen in terms of costs, with a 44% lower average, the LCA reveals that gray hydrogen exhibits a GWP100 impact about 2.5 times higher than green hydrogen on average. Overall, fuel usage has the highest impact on both costs and GWP100, therefore showing that lower speeds and thus lower fuel usage make the AHB perform better. The hull has a larger impact in both categories early on (19% of costs and 70% of GWP100 in case of green hydrogen and 24% of costs and 48% of GWP100 in case of gray hydrogen) but decreases in importance as speeds increase and more kilometers per year are travelled. Loading has an especially large impact on costs (9% to 30% of costs for green hydrogen and 21% to 39% of costs for gray hydrogen) but is negligible for GWP100. All other parameters increase with speed.

Next to the influence of all boat specific factors on the TCO, it is important to also consider the impact of the carbon price. This regulatory mechanism, such as the European Emissions Trading System (EU ETS)[29], requires users to pay for their carbon emissions, thereby favoring cleaner fuels like green hydrogen. With an average projected EU ETS carbon price of 95 € ton$^{-1}$ CO$_2$ eq. until 2030[30], it becomes evident that operating the AHB with gray hydrogen will incur higher average additional costs (0.10 € km$^{-1}$) compared to using green hydrogen (0.04 € km$^{-1}$). While introducing carbon pricing is a way to move towards a more sustainable economy, the TCO calculations show that the AHB operated by gray hydrogen is still much cheaper, requiring even higher carbon prices to reach a break-even point.

Apart from the carbon price, decreasing prices for green hydrogen could also lead to a reduction in the AHB's TCO. Since they are expected to decrease rapidly over the coming decades (from 7 € kg$^{-1}$ to around 3 € kg$^{-1}$)[31], red error bars have been added to Fig. 3 to illustrate the potential impact of green hydrogen price decreases on the TCO. When the green hydrogen price is set to 3 € kg$^{-1}$ across all scenarios, the average TCO of the AHB, excluding the carbon price, is 0.50 € km$^{-1}$. This is significantly lower than the base case with a green hydrogen price of 7 € kg$^{-1}$ (0.82 € km$^{-1}$). No error bars are shown for gray hydrogen, as its price is expected to remain stable[31].

Going back to the four base scenarios, one exception of a general increase in costs or environmental impact when going faster is the GWP100 value of green hydrogen at 10 km h$^{-1}$. Compared to all other scenarios, the value is the lowest here. This is attributed to the superior performance per kilometer of the fixed components of the AHB, without yet reaching a fuel usage increase high enough to entirely offset the fixed components. Because scenario 2 has the lowest GWP100 value compared to the other scenarios, an adequate TCO, and the literature values for the reference fuel cell electric truck[26] are based on green hydrogen, this scenario is compared to the three reference trucks in the next section. As for Fig. 3, it can lastly be seen that the AHB powered by gray hydrogen outperforms all reference scenarios when looking at costs while the AHB powered by green hydrogen is only the cheapest option for scenarios 1 and 2.

A marginal abatement cost analysis for the AHB in scenario 2, powered by either green or gray hydrogen, compared to a reference diesel truck (TCO only)[26] and considering average CO$_2$ emissions of diesel trucks (type 5-LH) in the EU28[32] reveals the following: The AHB powered by green hydrogen has a marginal abatement cost of -0.17 € kg$^{-1}$ CO$_2$ eq compared to the diesel truck. In contrast, the AHB powered by gray hydrogen demonstrates a substantially more negative marginal abatement cost of -4.64 € kg$^{-1}$ CO$_2$ eq. While the highly negative marginal abatement costs for gray hydrogen powered AHBs appear economically and ecologically advantageous, they do not reflect reality. This is because reducing the carbon footprint of gray hydrogen would effectively alter its classification, leading to entirely different cost structures[31] (see supplementary note 3 in the supplementary materials for more details).

Besides green and gray hydrogen, blue hydrogen is another option. Like gray hydrogen, blue hydrogen is produced through steam reforming of natural gas, but it additionally incorporates carbon capture with efficiencies ranging from 50% to 95%[33], thereby reducing greenhouse gas emissions. With an expected future price of around 2 € kg$^{-1}$ [31], the TCO for AHBs using blue hydrogen could match that of gray hydrogen if the carbon price is excluded. However, when the carbon price is included, blue hydrogen offers superior performance due to its reduced emissions. Due to the emission reduction, the AHB's GWP100 per km also improves, ranging from 0.64 kg CO$_2$ eq km$^{-1}$ at 50% carbon capture efficiency to 0.22 kg CO$_2$ eq km$^{-1}$ at 95% efficiency. These findings suggest that high carbon capture efficiencies may rival or exceed the outcomes with green hydrogen. Despite the apparent advantages of blue hydrogen, it should be noted that these calculations assume no additional emissions from the auxiliary systems required for carbon capture. Including these in a comprehensive LCA would likely result in less favorable outcomes.

Since the model AHB is based on several assumptions, a sensitivity analysis is carried out. This sheds light on important parameters and reveals how a change in base values can alter the overall TCO. Results for the sensitivity analysis of scenario 2 for green hydrogen are illustrated in Fig. 4. They show that a change of the total drag coefficient, the wetted surface area, or the hydrogen cost has the largest influence on the TCO. Given that variations in hydrogen costs have thoroughly been accounted for in our analysis of current and future costs for green and gray hydrogen, along with the extensive discourse on future hydrogen costs in the existing literature[31,34,35], we are confident that our assumption is adequately substantiated. As discussed above, the wetted surface area and the drag coefficient were derived utilizing an approach by Molland[36] (see supplementary note 1 of the supplementary materials).

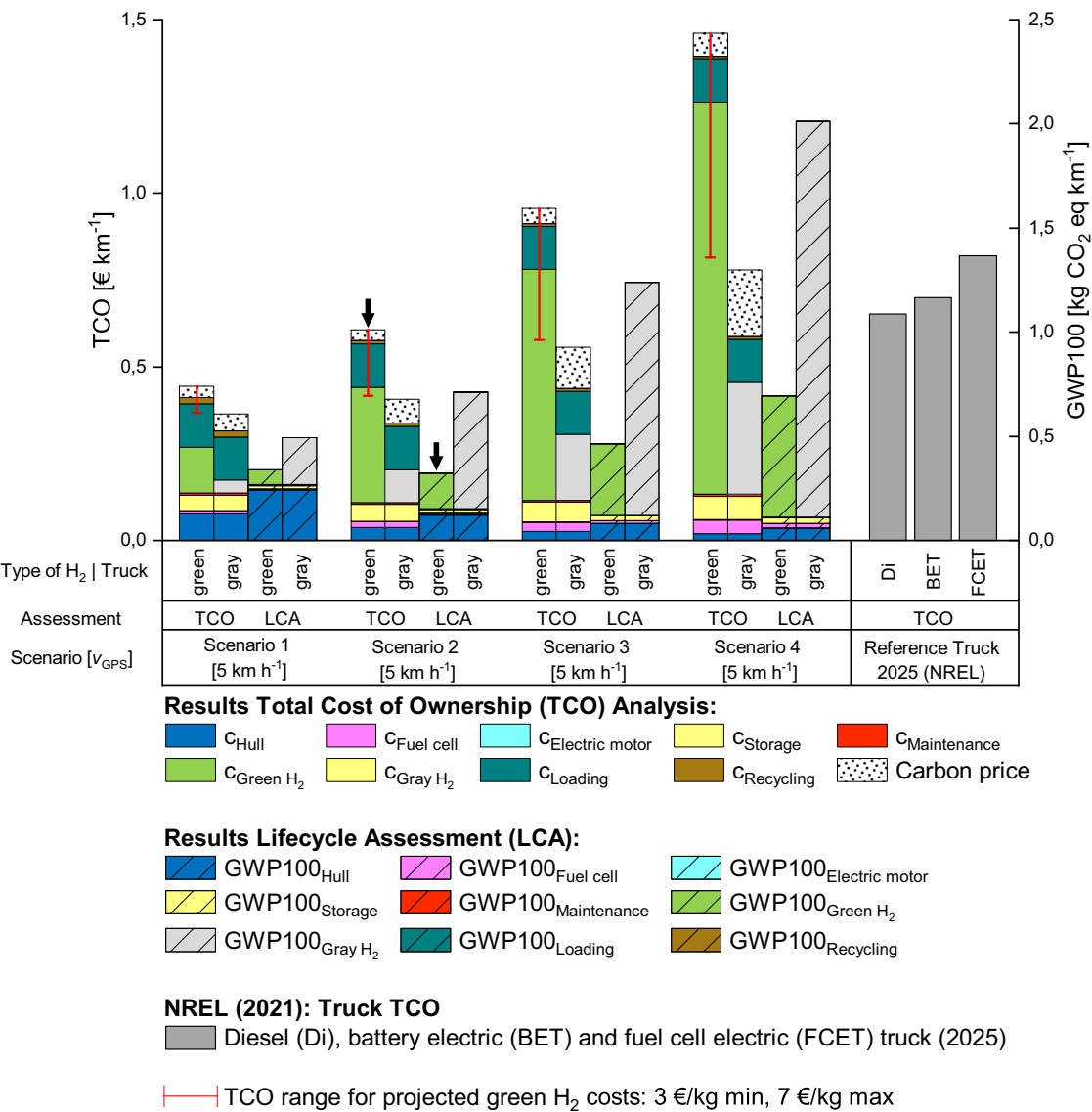

**Fig. 3 | Four scenarios based on the speed of the autonomous-driving hydrogen-powered boat (AHB) featuring the results of the combined TCO analysis and LCA for green and gray hydrogen broken down into the respective parameters.** The three gray bars on the right are TCO values taken from literature[26]. They represent the costs for owning and operating three different types of trucks (diesel, battery-electric (BET), and fuel-cell electric (FCET)) in the year 2025 and are used for comparison to the model AHB. Error bars show the TCO price range based on expected green hydrogen prices[31]. The bottom end of the error bars represents the TCO of the respective scenario when the price for green hydrogen is set to 3 € kg$^{-1}$, while the top end of the error bars reflects a green hydrogen price of 7 € kg$^{-1}$. Black arrows indicate the scenario utilized for further analysis.

## Comparison of model AHB to different trucks

The aim of this research is not only to analyze the AHB's environmental and economic performance but to also compare AHBs to different kinds of trucks proposed in prior literature[26]. Since the model AHB incorporates the most optimal tradeoff between costs and GWP100 in scenario 2 (green hydrogen), this scenario is chosen for further analysis. The results, delineating the direct TCO comparison between the reference trucks and the model AHB, with a specific emphasis on the European context, are depicted in Fig. 5.

As a first step, the total trucking market of the EU27 countries[37] is split into the two categories time sensitive (e.g., agriculture and food products) and not time sensitive (e.g., coal, metal ores). This is done by utilizing the standard goods classification for transport statistics[38] provided within the market dataset. Table S8 shows the result of the clustering. Within these clusters, it is then possible to directly allocate part of the market into three

distance segments (1 to 499 km per trip, 500 to 999 km per trip, 1000+ km per trip) which is indicated by the six pie diagrams (see Table S9).

Finally, a cost assessment for all distances is carried out for both the AHB and the semi-trucks (see Tables S8 to S10). As illustrated in Fig. 5, the model AHB is not the preferred solution for the time sensitive segment. This can be attributed to the lower speed of the AHB relative to trucks, which typically travel at much higher speeds[39]. Consequently, when time is critical, goods will always be transported by trucks. For non-time sensitive goods, the speed limitation does not apply and the TCO per kilometer is the only relevant measurement.

Assuming exclusive diesel truck usage in future truck transport, the AHB could potentially replace diesel trucks for trips exceeding 576 km due to its lower TCO. If future truck transport shifts entirely to vehicles powered by batteries or hydrogen fuel cells, the AHB could substitute these vehicles for journeys of 624 km and longer. In this case, the AHB would represent

**Fig. 4 | Sensitivity analysis for scenario 2 (green hydrogen) of the total cost of ownership analysis.** All base values for parameters were varied by 50% in either direction while other parameters remained at base level. Exceptions are base values for the efficiency of the fuel cell and the electric motor to better reflect feasible values.

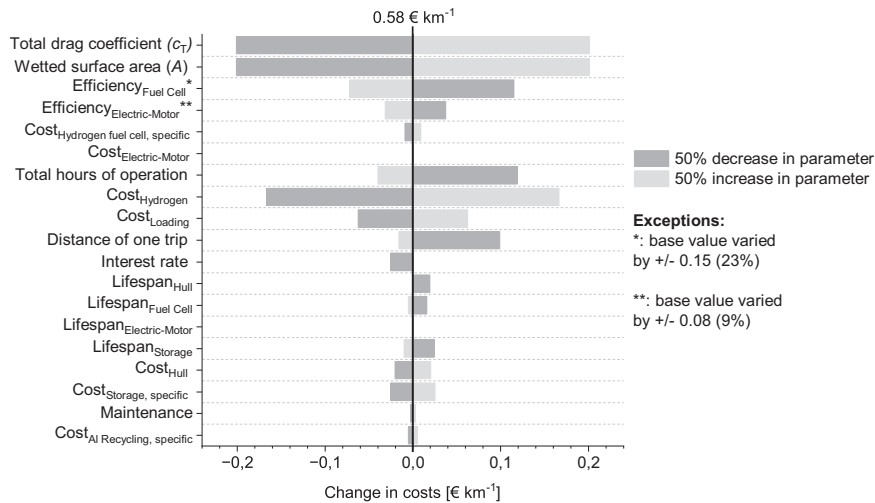

18.21% of the European trucking market as the most cost-effective transport alternative to non-time-sensitive long-distance trucking.

### Potential market share of the AHB

The results of Fig. 5 can be translated into a potential market outlook for the field of intracontinental container freight transport of the EU27 countries which per our definition includes the market segments of trucking, trains, and inland waterway transport. Assuming that the AHB will capture 18.21% of the trucking market, Fig. 6 provides a market projection for the year 2030. To calculate the market projection, compound annual growth rates for each market segment were taken from literature[40–42]. This suggests that AHBs can be a viable alternative and could, in theory, take over a sizable portion of the European freight transport market. With expected decreases in green hydrogen price[31], it is possible that the AHB could perform even better when compared to long-haul trucks in the future. However, as trucking will also keep evolving in regard to battery and fuel cell technology, future research may update our calculations when technological choices and prices become clearer.

### Conclusion

By modelling the design parameters of an AHB and then carrying out a combined environmental and cost analysis, this research offers several insights. First, results of this study suggest that AHBs can, due to their smaller size, expand the accessible river network and therefore potentially substitute emission-intensive truck transportation. Second, this study determines the TCO and environmental impact measured in GWP100 for an AHB, thus providing first insights into the environmental and economic potential of AHBs. Focusing on scenario 2 and green hydrogen, the results suggest that a cruising speed of 10 km h$^{-1}$ is preferred because it incorporates a great balance between GWP100 (0.33 kg CO$_2$ eq km$^{-1}$) and TCO (0.58 € km$^{-1}$). Third, the study expands prior literature by comparing the costs of the modelled AHB with different future semi-trucks. The findings suggest that compared to trucks, the AHB is the cost-optimal option for non-time sensitive intracontinental container transport for distances above 624 km. Finally, this study provides an outlook for the potential intracontinental freight transport market with AHBs in 2030. This demonstrates that the AHB could take up a noteworthy market share in the future.

This study is a first attempt at analyzing the environmental and economic potential of AHBs. As such it is not without limitations, but these limitations provide opportunities for future research. While the combined environmental and cost analysis offers new insights, it is based

on several assumptions. Incorporating primary data of the AHB, with actual measurements of the drag coefficient (including air drag) and the wetted surface area, would improve the accuracy of the calculations. Additionally, a more intricate modeling of the AHB, integrating a battery for power bursts and additional auxiliaries, could further refine the analysis.

Future research could also contrast the AHB to other modes of transportation like trains, airplanes, and inland waterway transport. Additionally, examining the environmental impact of the AHB in relation to those of semi-trucks could offer another research opportunity, further contributing to the discourse on sustainable transportation solutions. Finally, future research could consider new legal frameworks and costs needed to officially register AHBs.

Overall, AHBs could be a promising way to re-think the transportation sector and could make the intracontinental container freight transport more sustainable.

### Methods

This study features a combined cost and life cycle assessment for modelling an autonomous-driving hydrogen-powered river freight boat capable of transporting one forty-foot-equivalent-unit container on inland waterways. For clarification, Fig. 2, provided the model design of the AHB and its most important design parameters.

### Cost assessment

The cost assessment is carried out via a high-level process-based cost model (PBCM) approach, similar to earlier work[43,44]. The PBCM was deemed as a fitting approach because it calculates costs based on technical parameters from the bottom up, thus breaking down a complex and so far under-explored problem into smaller steps. On the one hand, this is especially beneficial as it allows for the use of established technical data instead of lacking real-world data. On the other hand, it offers transparency by showing which parameters contribute the most to total costs[45].

To make results between the cost assessment and LCA comparable, the total costs are levelized to costs per km and year (LCPK). Because the AHB is modelled to have transport capabilities of exactly one FEU container, the LCPK is also automatically levelized to one container. As shown in Eq. (1), the LCPK of the AHB is, in principle, calculated by dividing the sum of the annualized capital expenditures (CAPEX$_{Annualized}$), the annual operational expenditures (OPEX), and the annualized recycling costs (Recycling$_{Annualized}$) by the amount of km travelled per year. Additionally, the LCPK is adjusted using a fixed maintenance rate which is assumed to be a percentage of the overall

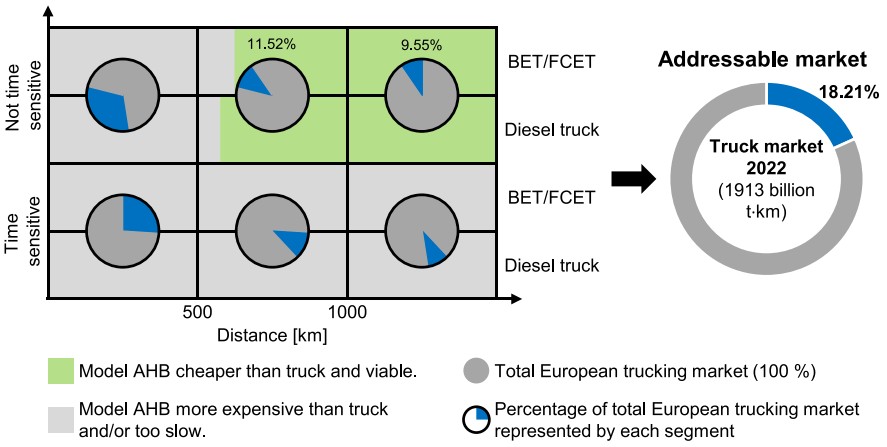

**Fig. 5 | Comparison of total cost of ownership for model autonomous-driving hydrogen-powered boat (AHB) against battery-electric (BET), fuel-cell electric (FCET), and diesel trucks across different trip distances.** The least expensive option is selected between BET and FCET. AHB data is derived from scenario 2 (10 km h⁻¹, green hydrogen). Truck data is taken from literature[26]. Pie charts detail the EU trucking market distribution, categorized by time-sensitive and non-time sensitive goods according to standard goods classification for transport statistics[38]. The AHB reaches the break-even point with diesel trucks at 576 km and at 624 km for BET/FCET. The right panel shows the AHB's market potential based on the BET/FCET break-even point.

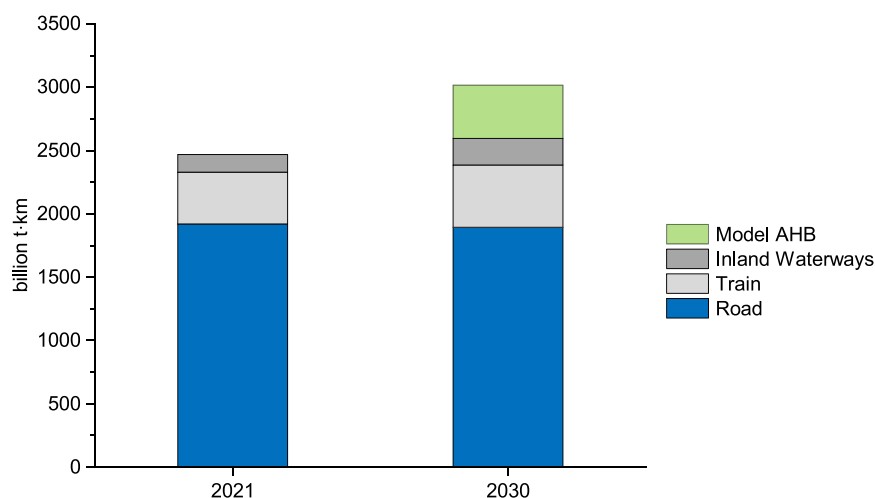

**Fig. 6 | Potential market projection for the intra-continental container freight transport of the EU27 countries.** The model autonomous-driving hydrogen-powered boat (AHB) replaces 18.21% of the trucking market. Data for the EU27 intracontinental container freight transport market in 2021 is taken from eurostat[37]. The market size in 2030 is based on expected compound annual growth rates (CAGR) of each subcategory. CAGR for European truck freight transport (2022 to 2027): 2.1%[42]; CAGR for global train freight transport (2020-2026): 2%[41]; CAGR for European inland waterways freight transport (until 2028): 5%[40].

CAPEX. Similar approaches to estimate the maintenance costs for PBCMs have previously been applied in literature and are therefore adopted by this work[45–49].

$$\mathrm{LCPK} = \frac{\mathrm{CAPEX_{Annualized}}(1 + \text{maintenance rate}) + \mathrm{OPEX} + \mathrm{Recycling_{Annualized}}}{\text{km per year}}$$
(1)

The annualized CAPEX (Eq. (2)) is made up of four distinct cost factors which include the costs for the hull, the fuel cell stack, the electric motor, and hydrogen storage. To ensure that capital costs (borrowing money at an interest rate $i$) are included, the four CAPEX cost factors are multiplied by a factor-specific capital recovery factor $crf_j$ (Eq. (3)), thus resulting in an annualized CAPEX. The factor specific capital recovery factor varies depending on the assumed lifespan of the CAPEX cost factor $j$.

$$\begin{aligned}\mathrm{CAPEX_{annualized}} = {} & crf_{\mathrm{Hull}} \cdot c_{\mathrm{Hull}} + crf_{\mathrm{Fuel\,cell\,stack}} \cdot c_{\mathrm{Fuel\,cell\,stack}} \\ & + crf_{\mathrm{Electric\,motor}} \cdot c_{\mathrm{Electric\,motor}} + crf_{\mathrm{Storage}} \cdot c_{\mathrm{Storage}}\end{aligned}$$
(2)

$$crf_j = \frac{i(1 + i)^{\mathrm{Lifespan}_j}}{(1 + i)^{\mathrm{Lifespan}_j} - 1}$$
(3)

While the costs of the hull are assumed to be of a fixed value, the costs for the fuel cell stack, the electric motor and storage are influenced by more underlying parameters.

The costs for a suitably powerful hydrogen fuel cell stack can be derived using Eq. (4), in which the required power of the hydrogen fuel cell stack (calculated in Eq. (5)) to sustain the cruise speed $v_{\mathrm{R,\,max}}$ (maximum relative speed of AHB compared to the river flow) is multiplied by the fixed specific costs of a hydrogen fuel cell. Equation (5) is of upmost importance to the PBCM, as it determines the power needed to move the boat forward at a sustained cruise speed. It achieves this by calculating the drag force of water imposed on the moving AHB ($F_{\mathrm{Drag}}$), which can be calculated using the drag formula, and multiplying it by the AHB's relative speed compared to the water $v_R$. Here, $\rho$ stands for the mass density of water, $c_T$ represents the total drag coefficient and $A$ the wetted surface area of the AHB. We decided to leave out other forces enacted on the AHB like the air drag to reduce the complexity of the model.

$$c_{\mathrm{Fuel\,cell\,stack}} = \mathrm{Power_{Fuel\,cell\,stack;\,sustain\,cruise\,speed}}(v_{\mathrm{R,max}}) \cdot c_{\mathrm{Fuel\,cell\,stack,specific}}$$
(4)

$$\mathrm{Power_{Fuel\,Cell\,Stack;\,Sustain\,Cruise\,Speed}} = F_{\mathrm{Drag,boat}} \cdot v_R = 0.5 \cdot \rho \cdot v_R^3 \cdot c_T \cdot A$$
(5)

The costs for an electric motor of fitting power are determined by multiplying the power needed at $v_{\mathrm{R,\,max}}$ with the fixed specific costs of

electric motors (Eq. (6)).

$$c_{\text{Electric motor}} = \text{Power}_{\text{Fuel cell stack; sustain cruise speed}}(v_{R,\text{max}}) \cdot c_{\text{Electric motor,specific}}$$

$$(6)$$

The costs for storage of hydrogen are calculated using Eq. (7) in which the fixed specific storage costs for hydrogen are multiplied by the necessary storage capacity of hydrogen. In a last step to finalize the calculation of all CAPEX-relevant parameters, the storage costs of hydrogen are determined using Eq. (8). Here, the quotient of the average distance of the trip and the maximum speed of the AHB (which will lead to maximum fuel consumption) is multiplied with the power necessary to achieve maximum speed. Furthermore, this result is multiplied with the inverse of the efficiencies of the fuel cells and the electric motor to account for energy losses of the complete drivetrain, from hydrogen gas to movement of the propeller. To model the possibility of storing a small surplus of hydrogen, the intermediary result is multiplied by a constant of 1.1.

$$c_{\text{Storage,total}} = c_{\text{Storage,specific}} \cdot \text{Hydrogen}_{\text{Storage capacity}}(v_{R,\text{max}}) \qquad (7)$$

$$\text{Hydrogen}_{\text{Storage capacity}} = \left( \frac{\text{Distance}_{\text{Trip}}}{v_{R,\text{max}}} \cdot \text{Power}_{\text{Fuel cell stack}}(v_{R,\text{max}}) \right.$$
$$\left. \cdot \frac{1}{\text{Efficiency}_{\text{Fuel cell stack}}} \cdot \frac{1}{\text{Efficiency}_{\text{Electric motor}}} \right) \cdot 1.1$$

$$(8)$$

The annual OPEX (Eq. (9)) is calculated by multiplying the fuel consumption per km (which depends on $v_R$) with the cost of hydrogen per kg as well as km travelled per year and then adding the total loading costs.

$$\text{OPEX} = \text{Fuel consumption per km}(v_R) \cdot c_{\text{Hydrogen}} \cdot \text{km per year} + c_{\text{Loading,total}}$$

$$(9)$$

Equation (10) shows the calculation of the fuel consumption per km, wherein the energy requirement of the hydrogen fuel cell per hour is first divided by the product of $v_R$ and time $t$. This is then multiplied with the inverse of the lower heating value of hydrogen and the efficiencies of the fuel cells and the electric motor. The resulting overall drop in efficiency once again accounts for the energy losses of the drivetrain.

$$\text{Fuel Consumption per km}(v_R) = \frac{E_{\text{Fuel cell stack}}(v_R)}{v_R \cdot t} \cdot \frac{1}{LHV_{\text{Hydrogen}}}$$
$$\cdot \frac{1}{\text{Efficiency}_{\text{Fuel cell stack}}} \cdot \frac{1}{\text{Efficiency}_{\text{Electric motor}}}$$

$$(10)$$

The kilometers travelled per year are calculated by multiplying the GPS speed of the AHB ($v_{GPS}$ is determined by the sum of $v_R$ and the speed of the river flow, it is assumed to be constant against and with the river flow which means that $v_R$ has to change depending on the direction the AHB is travelling. This is why for some calculations, $v_{R,\text{max}}$ is used) with the total hours of operation per year (Eq. (11)).

$$\text{km per year} = v_{\text{GPS}} \cdot \text{total hours of operation} \qquad (11)$$

The total loading costs (estimation of staff costs and port fees) per year are determined by multiplying the fixed loading costs per trip with the ratio of km travelled per year and average distance per trip (see Eq. (12)).

$$c_{\text{Loading,total}} = c_{\text{Loading per trip}} \cdot \text{Loading frequency per year}$$
$$= c_{\text{Loading per trip}} \cdot \frac{\text{km per year}}{\text{Distance}_{\text{Trip}}}$$

$$(12)$$

Finally, the annualized recycling costs are calculated utilizing Eq. (13), where the specific costs for the recycling of aluminum per kg are multiplied by the total weight of end-of-life boat scrap and then divided by the lifespan of the boat. To reduce complexity, the scrap is assumed to be aluminum scrap only.

$$\text{Recycling}_{\text{Annualized}} = \frac{c_{\text{Al recycling,specific}} \cdot \text{Scrap weight}_{\text{Total}}}{\text{Lifespan}_{\text{Boat}}} \qquad (13)$$

An overview of all input parameters for Eq. (1) to Eq. (13) can be found in Table S7 of the supplementary materials.

### Sensitivity analysis

Given the study's emphasis on assessing the cost competitiveness of the AHB in comparison to various truck types and recognizing the substantial variation in total cost of ownership based on selected parameter values, it becomes imperative to address uncertainty surrounding these key parameters. Therefore, a sensitivity analysis is carried out in which the TCO model parameters shown in Table S7 of the supplementary materials are varied by 50% in either direction. Only the parameters for fuel cell and electric motor efficiency have been varied less as this would lead to unrealistic values. As a result, it can effectively be shown which parameters have the largest influence on the TCO outcome and therefore need to be chosen most carefully.

### Life cycle assessment

Life cycle assessments carried out within a standardized framework are a great tool for analyzing and calculating environmental impacts of any project of choice. As stated in ISO14040/14044[50,51], the standardized approach for conducting LCAs consists of the four following steps: 1. Defining the goal and scope of the LCA, 2. Setting up a life cycle inventory (LCI), 3. Conducting a life cycle impact assessment (LCIA) and 4. Interpreting the results. Therefore, all these steps will be touched on subsequently. Because our approach features a combined cost and life cycle assessment, the structure of the LCA will be aligned with the previously described structure of the cost assessment.

The goal of this LCA is to analyze the environmental impact of manufacturing, operating, and finally disposing of the proposed theoretical AHB at its end-of-life. The system boundary can therefore be defined as being cradle-to-grave. The goal will be achieved via an attributional cut-off LCA that focuses on the direct physical flows (raw materials, energy and emissions) of a product across its relevant lifetime[52,53]. Within LCAs, it is important to select a functional unit which describes the quantifiable product or service that serves as a reference throughout the analysis[50,51]. To stay consistent with the cost assessment, a functional unit of 1 km of AHB travel is chosen.

The life cycle inventory provides data about raw material, energy and waste flows necessary for the project of interest during the relevant stages of its life cycle. This usually includes production, operation and end-of-life or in other words, all activities from cradle-to-grave[54]. For this paper, data was primarily collected from academic literature and the well-known environmental database Ecoinvent 3.9.1[52,55]. To simplify the implementation of the LCA, the open-source software openLCA 2.0.3 was utilized[56,57]. More information regarding the life cycle inventory can be found in supplementary note 6 of the supplementary materials.

During the life cycle impact assessment, data from the LCI (material, energy and waste flows) is converted into a more compressed set of environmental impact categories using different established LCIA characterization methods. In this study, the Environmental Footprint (EF) 3.1[58] was employed which condenses results into 16 impact categories like ecotoxicity for freshwater, human carcinogenic toxicity, acidification, and climate change in the form of global warming potential (GWP). While all these categories provide insightful results, literature and politics have often focused strongly on the impact category climate change, e.g., in the form of carbon footprints, because of its pressing relevance[54,59–61]. This is why the

focus of this study is on this impact category as well. *GWP100* is calculated utilizing EF 3.1 and measures climate change caused by greenhouse gases over the next 100 years. It does this by converting and accumulating the emissions of greenhouse gases like $CO_2$, $CH_4$ or $NO_x$ into so-called $CO_2$ equivalents that might arise during the production, usage and end-of-life stage of the product or service[43]. In this work, scope 1, 2, and 3 emissions were considered. While scope 1 and 2 emissions focus on the direct and indirect greenhouse gas emissions of a product, scope 3 emissions take all indirect emissions up and down the value chain of the product into account[62]. For the sake of completeness, Tables S2-S4 of supplementary note 2 provide a comprehensive overview of the results for all impact categories.

To be able to interpret the *GWP100* results derived from the LCIA and compare them to the TCO-model, the results are converted to the functional unit of 1 km of AHB travel utilizing the following equations. Wherever applicable, parameter values are kept consistent when compared to the TCO analysis. The overall *GWP100* result per km (Eq. (14)) is calculated by summing the fixed, variable, and recycling global warming potentials and then dividing this sum by the total kilometers travelled per year. Again, this study included a maintenance parameter set to 0.05 to stay consistent with the TCO analysis.

$$GWP100 \text{ result per km}$$
$$= \frac{GWP100_{\text{fixed}}(1 + \text{maintenance}) + GWP100_{\text{variable}} + GWP100_{Recycling}}{\text{km per year}}$$
(14)

The fixed *GWP100* result is determined by multiplying the *GWP100* of each fixed AHB component with its respective impact recovery factor $irf_j$ and then summing all intermediary results (Eq. (15)).

$$GWP100_{\text{fixed}} \text{result} = irf_{\text{Hull}} \cdot GWP100_{\text{Hull}} + irf_{\text{Fuel cell stack}} \cdot GWP100_{\text{Fuel cell stack}}$$
$$+ irf_{\text{Electric motor}} \cdot GWP100_{\text{Electric motor}}$$
$$+ irf_{\text{Storage tank}} \cdot GWP100_{\text{Storage tank}}$$
(15)

Since interest rates do not exist for calculating environmental impacts of fixed parameters, this study simplifies *crf* as used in Eq. (3) to *irf* (Eq. (16)).

$$irf_j = \frac{1}{\text{Lifespan}_j}$$
(16)

While the $GWP100_{\text{Hull}}$ can be determined directly in openLCA because it is modelled utilizing the hull's final measurements, the *GWP100* of the other fixed AHB components needs to be broken down further. As shown in Eq. (17), $GWP100_{\text{Fuel cell stack}}$ is calculated by multiplying unit_$GWP100_{\text{Fuel cell stack}}$ with the required power of the fuel cell stack at each respective $v_{R, \text{max}}$. Here, unit_$GWP100_{\text{Fuel cell stack}}$ stands for the global warming potential caused by a fuel cell stack with 1 kW power. The value of this parameter originates from the LCIA conducted in openLCA.

$$GWP100_{\text{Fuel cell stack}} = unit\_GWP100_{\text{Fuel cell stack}}$$
$$\cdot \text{Power}_{\text{Fuel cell stack}}(v_{R,\text{max}})$$
(17)

To determine $GWP100_{\text{Electric motor}}$, the global warming potential per kW of electric motor is once again multiplied by the power requirements of the electric motor at $v_{R, \text{max}}$ (Eq. (18)).

$$GWP100_{\text{Electric motor}} = unit\_GWP100_{\text{Electric motor}} \cdot \text{Power}_{\text{Electric Motor}}(v_{R,\text{max}})$$
(18)

Accordingly, $GWP100_{\text{Storage tank}}$ can be calculated by multiplying the global warming potential caused per kg of Hydrogen with the required hydrogen storage capacity as shown in Eq. (19).

$$GWP100_{\text{Storage tank}} = unit\_GWP100_{\text{Storage tank}} \cdot \text{Hydrogen}_{\text{Storage capacity}}$$
(19)

The variable *GWP100* result is determined utilizing Eq. (20), whereby $GWP100_{\text{Hydrogen}}$ is added to $GWP100_{\text{Loading}}$.

$$GWP100_{\text{variable}}\text{result} = GWP100_{\text{Hydrogen}} + GWP100_{\text{Loading}}$$
(20)

$GWP100_{\text{Hydrogen}}$ is calculated by multiplying the fuel consumption per km at speed $v_R$ with the global warming potential of green or gray hydrogen per kg and the kilometers of AHB travel per year at speed $v_R$ (Eq. (21)).

$$GWP100_{\text{Hydrogen}} = \text{Fuel Consumption per km}(v_R)$$
$$\cdot unit\_GWP100_{\text{Hydrogen}} \cdot \text{km per year}(v_R)$$
(21)

$GWP100_{\text{Loading}}$ is determined utilizing Eq. (22), where the unit *GWP100* impact of loading per ton and kilometer is multiplied by the weight of the freight and the km travelled per year.

$$GWP100_{\text{Loading}} = unit\_GWP100_{\text{Loading per ton and km}} \cdot \text{weight}_{\text{freight}} \cdot \text{km per year}$$
(22)

Finally, $GWP100_{\text{Recycling}}$ is calculated by multiplying the unit *GWP100* impact of aluminum recycling per kg by the AHB's end-of-life scrap weight and then dividing it by the total lifespan of the boat. To stay consistent, all scrap is assumed to be aluminum scrap.

$$GWP100_{\text{Recycling}} = \frac{unit\_GWP100_{\text{Recycling}} \cdot \text{Scrap weight}_{\text{Total}}}{\text{Lifespan}_{\text{Boat}}}$$
(23)

## Data availability
Data supporting the results of the present study are presented in the Supplementary Materials. LCI data as well as LCI data sources are provided in the Supplementary Materials. Data and sources for cost analysis are given in the Supplementary Materials.

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

## Author contributions

S.S. and M.G. conceived the idea. S.S. designed the research. S.S., M.G., N.K., and F.F. conceived the illustrations. S.S. performed the analysis. S.v.D. guided and supervised the project. S.S. and S.v.D. wrote and revised the manuscript. M.G., N.K., and F.F. reviewed the manuscript.

## Funding

## Competing interests

N.K. is an employee at Porsche Consulting GmbH. F.F. is an employee at FutureCamp Climate GmbH. The other authors declare no competing interests.
