## [Transparent Peer Review file · Communications Engineering]

Assessment of environmental impacts and costs for hydrogen-powered cargo shipping in Europe

Corresponding Author: Mr Simon Schlehuber

Version 0:

Reviewer comments:

Reviewer #1

(Remarks to the Author)

The manuscript describes an environmental and economic study on deploying autonomous-driving (sailing) hydrogen powered boats (AHBs) in the European context. The work includes a total cost of ownership analysis and partial life-cycle analysis. The study reported in this manuscript is original, and relevant, but I have some major comments for consideration, before the manuscript can be published:

I note that deterministic values were used for fuel prices. These were set at 2 EUR/kg for grey hydrogen and 7 EUR/kg for green hydrogen. It is clear that these prices will vary, and a probabilistic approach could have been useful in exploring this effect. I can see that the sensitivity analysis explores this to some extent, using the variation in prices, but I think that at least a comment on this should be added to clarify that prices are expected to fluctuate with seasons and geopolitical events. – Further to the higher-frequency variations in price mentioned so far, there is also expected to be a long-term development of the prices, based on technological development, climate negotiations etc. Has this been taken into account (since you are making projections for future transport (year 2023 in Figure 6))? I think the authors should at least discuss the long-term price trends for green and grey hydrogen, and mention that these are not necessarily expected to remain constant over time.

Grey hydrogen was used to compare green hydrogen with. My immediate question was: “What is the point of using grey hydrogen in the context of the current climate crisis, and the knowledge that the pressure to reach net-zero greenhouse gas emissions is already agreed upon in the Paris Agreement?”. This seems like an outdated approach. Why not consider blue hydrogen? Whilst I do not want to impose adding calculations for blue hydrogen, I think at least a brief discussion about this choice should be included in the article.

Did the article consider any cost for CO₂ emissions? It seems very surprising that despite values of EUR being used and the context being Europe (and more specifically EU countries), no costs are considered for the EU ETS system and the price of tonne of CO₂ emitted. This is very surprising, and I think a major shortcoming of this work. The EU ETS is expected to have a significant impact on the costs, and it will significantly influence the results for the use of grey hydrogen. In my opinion this discussion (ideally supported with some calculations) must be included in the article before publication. If this is supposed to be for a more general (non-EU context), then the cost per tonne of CO₂ necessary to make green and grey hydrogen cost-competitive with one another could be calculated, in order to provide some useful policy advice.

I could not follow the argumentation of why the “end-of-life” phase was ignored in the LCA, when this is clearly defined in ISO14040. To me this is like selling a car, but without any wheels or engine. Do you mean that you are going to let the AHBs float down the river when you are done with them? Isn't the whole point of doing an LCA to include all phases of the life-cycle? Why do we otherwise have an ISO standard on this if parts of it are going to be ignored? Of course you can buy your own engine and wheels for your car that you bought without them, but it seems like a very strange concept, and to the non-trained eye may provide distorted results with respect to a proper LCA. In my opinion the results should be revised by taking the end-of-life phase into account, and the discussion should be updated.

As a potential added value for this manuscript, it would be possible to combine the effects of costs and greenhouse gas emissions in a marginal abatement cost (cost for CO₂ reduction between various options). Can you calculate the marginal abatement cost with respect to road transport, for the green and grey hydrogen options? This could be an interesting value for the reader, and can put the implementation of AHBs into context with other green concepts/technologies.

Minor comments:

A clear overview of the various scenarios should be provided. For example: “The speeds considered [for each of the

scenarios] are 5 km/h, 10 km/h, 15 km/h, and 20 km/h and describe the speed of the AHB as seen on the global positioning system (GPS).”

“This, and the fact that the literature values for a fuel cell electric truck are based on green hydrogen are the reason why scenario 2 for green hydrogen is compared more closely to all three reference trucks in the next section.” This sentence was not clear to me, please elaborate to explain more clearly what was meant.

Reviewer #2

(Remarks to the Author)

The authors provide insight into the use of AHBs (autonomous, hydrogen-powered boats) as an alternative to long-haul trucks by comparing the costs and emissions of both options. Thus, the manuscript addresses an important topic, as long-haul trucking is challenging to make more sustainable, and the transport sector generally has high emissions. The comparison and choice of scenario result in a particularly useful contribution. However, there are a number of unclarities and remarks that should be addressed before publishing, in my opinion, detailed below.

- The transport sector is large and diverse (long-distance ships, planes, trucks, etc.), but this manuscript mainly focuses on long-haul trucking and alternatives. This should also be mentioned in the first paragraph of the manuscript (the amount of goods and the emissions they cause when being transported strongly depends on the type of transport).
- Only the benefits of AHBs are mentioned. I believe there are several downsides, such as, but not limited to, the unclarity of the legal framework. Please mention these downsides as well
- In the introduction, it remains unclear whether you compare the AHBs to conventional inland ships or long-haul trucking
- Figure 1 is tilted; please undo this
- a straightforward method is missing in this manuscript: The methods are described but in several different sections. The readability of the manuscript would be significantly enhanced by having one single method section (after the introduction and before the results).
- Operational expenditures are broken down into fuel costs and loading costs (p 4, line 88): Where is the cost of human involved? Even though the ship is autonomous, people will likely still be involved (also for legal reasons). How is this, as compared to the truck?
- Line 93: better = lower?
- Types of hydrogen: why only gray and green? What about the other colours (mainly blue)
- Line 101 (life cycle assessment): how do you compare CO₂eq/GWP100 to money?
- I am missing a short method section where you summarize what exactly you are doing. This may be a table, or a separate section, but it would be nice to immediately understand what is in the model and how everything is calculated. Can be quite compact I believe
- Why do you consider the speeds (5 -20km/h), and what are current speeds of e.g. inland ships?
- Line 367: Why is decreasing the maintenance parameter deemed to be more adequate?

Version 1:

Reviewer comments:

Reviewer #1

(Remarks to the Author)

I have reviewed the revised version of the manuscript, and I am impressed by the fact that the authors have addressed all comments comprehensively and in an appropriate manner. The main improvements included updated calculations regarding the variations in fuel costs, the inclusion of the influence of a carbon-pricing scheme, and updating the LCA to include the end-of-life stage. Based on these thorough and appropriate revision done by the author, and based on the relevance and value of the study, I would like to recommend publication of the updated manuscript.

Reviewer #2

(Remarks to the Author)

The authors have effectively addressed the concerns raised in the previous reviews. The revisions have significantly improved the clarity and impact of the work. In its current form, I believe the manuscript to be suitable for publication.

Dear Reviewers,
thank you for taking the time to review our manuscript and for providing constructive
feedback, which has greatly helped us further developing the manuscript.

Below, you find a point-by-point response to your comments.

Reviewer #1:

**Comment 1.1**

I note that deterministic values were used for fuel prices. These were set at 2 EUR/kg
for grey hydrogen and 7 EUR/kg for green hydrogen. It is clear that these prices will
vary, and a probabilistic approach could have been useful in exploring this effect. I can
see that the sensitivity analysis explores this to some extent, using the variation in
prices, but I think that at least a comment on this should be added to clarify that prices
are expected to fluctuate with seasons and geopolitical events. – Further to the higher-
frequency variations in price mentioned so far, there is also expected to be a long-term
development of the prices, based on technological development, climate negotiations
etc. Has this been taken into account (since you are making projections for future
transport (year 2023 in Figure 6))? I think the authors should at least discuss the long-
term price trends for green and grey hydrogen, and mention that these are not
necessarily expected to remain constant over time.

**Response 1.1**

We agree with you that deterministic values for hydrogen prices do not reflect the
expected developments in the future, a fact that the provided sensitivity analysis could
not fully address. We have therefore decided to add an additional layer of information
to Figure 3 that shows the potential variation in the TCO of the AHB based on expected
future green and gray hydrogen prices.

As you correctly stated, the price for green hydrogen is expected to drop sharply; from
around 7 EUR/kg to below 3 EUR/kg by 2050¹. We have therefore added red error
bars in Figure 3 that indicate the resulting TCO range when taking projected green
hydrogen prices into account. Since the price is not expected to rise above 7 EUR/kg
(due to the rapid advancements in electrolysis technology), we have chosen to set 7
31 EUR/kg as the most conservative price (the one we are using for comparison to the
32 trucks in all further analyses) and 3 EUR/kg as the lowest price.

As seen in Figure 3, there are no errors bars for the TCO of the AHB powered by gray
hydrogen. This is due to the lack of expected price variation for gray hydrogen in the
future. According to literature, the price of gray hydrogen is expected to stay constant
at around 2 EUR/kg¹.

Below, you can find the specific changes to the manuscript as discussed in this section
(lines 160-167):

**Apart from the carbon price, decreasing prices for green hydrogen could also lead to**
**a reduction in the AHB's TCO. Since they are expected to decrease rapidly over the**
**coming decades (from 7 €/kg to around 3 €/kg), red error bars have been added to**

Figure 1: Four scenarios based on the speed of the AHB featuring the results of the combined TCO analysis and LCA for green and gray hydrogen broken down into the respective parameters. The three gray bars on the right are TCO values taken from literature²⁶. They represent the costs for owning and operating three different types of trucks (diesel, battery-electric (BET), and fuel-cell electric (FCET)) in the year 2025 and are used for comparison to the model AHB. Error bars show the TCO price range based on expected green hydrogen prices³¹. Black arrows indicate the scenario utilized for further analysis.

Figure 3 to illustrate the potential impact of green hydrogen price decreases on the
 TCO. When the green hydrogen price is set to 3 €/kg across all scenarios, the average
 TCO of the AHB, excluding the carbon price, is 0.50 €/km. This is significantly lower
 than the base case with a green hydrogen price of 7 €/kg (0.82 €/kg). No error bars
 are shown for gray hydrogen, as its price is expected to remain stable.

Furthermore, we have added the following to the discussion of Figure 6 (lines 252-
 256):

With expected decreases in green hydrogen price, it is possible that the AHB could
 perform even better when compared to long-haul trucks in the future. However, as
 trucking will also keep evolving in regard to battery and fuel cell technology, future
 research may update our calculations when technological choices and prices become
 clearer.

**Comment 1.2**

Grey hydrogen was used to compare green hydrogen with. My immediate question
 was: “What is the point of using grey hydrogen in the context of the current climate
 crisis, and the knowledge that the pressure to reach net-zero greenhouse gas
 emissions is already agreed upon in the Paris Agreement?”. This seems like an
 outdated approach. Why not consider blue hydrogen? Whilst I do not want to impose
 adding calculations for blue hydrogen, I think at least a brief discussion about this
 choice should be included in the article.

**Response 1.2**

We had initially decided to use gray instead of blue hydrogen as it is currently the
 dominant type of hydrogen on the market², and the comparison of the AHB versus the

trucks is based on expected data for 2025. Nevertheless, we agree with you that it
makes sense to additionally consider blue hydrogen because of the developments
toward clean energy. To make sure that Figure 3 does not become even more
complicated, we have decided to only discuss the results of the analysis for blue
hydrogen in the text of the manuscript.

The price for blue hydrogen is also expected to stay relatively constant at around
2 EUR/kg¹. While carbon capture technology is expected to mature and therefore
become cheaper, natural gas prices are expected to rise, therefore counteracting any
potential price savings. However, when taking carbon prices into account, blue
hydrogen is cheaper than gray hydrogen. The GWP100 range provided in the
manuscript is due to the difference in expected carbon capture efficiency which ranges
from 50 to 95%³.

Below, you can find the specific changes to the manuscript as discussed in this section
(lines 191-204):

Besides green and gray hydrogen, blue hydrogen is another option. Like gray
hydrogen, blue hydrogen is produced through steam reforming of natural gas, but it
additionally incorporates carbon capture with efficiencies ranging from 50% to 95%,
thereby reducing greenhouse gas emissions. With an expected future price of around
2 €/kg, the TCO for AHBs using blue hydrogen could match that of gray hydrogen if
the carbon price is excluded. However, when the carbon price is included, blue
hydrogen offers superior performance due to its reduced emissions. Due to the
emission reduction, the AHB's GWP100 per km also improves, ranging from 0.64 kg
CO₂ eq/km at 50% carbon capture efficiency to 0.22 kg CO₂ eq/km at 95% efficiency.
These findings suggest that high carbon capture efficiencies may rival or exceed the
outcomes with green hydrogen. Despite the apparent advantages of blue hydrogen, it
should be noted that these calculations assume no additional emissions from the
auxiliary systems required for carbon capture. Including these in a comprehensive LCA
would likely result in less favorable outcomes.

**Comment 1.3**

Did the article consider any cost for CO₂ emissions? It seems very surprising that
despite values of EUR being used and the context being Europe (and more specifically
EU countries), no costs are considered for the EU ETS system and the price of tonne
of CO₂ emitted. This is very surprising, and I think a major shortcoming of this work.
The EU ETS is expected to have a significant impact on the costs, and it will
significantly influence the results for the use of grey hydrogen. In my opinion this
discussion (ideally supported with some calculations) must be included in the article
before publication. If this is supposed to be for a more general (non-EU context), then
the cost per tonne of CO₂ necessary to make green and grey hydrogen cost-
competitive with one another could be calculated, in order to provide some useful policy
advice.

**Response 1.3**

We have now included the average projected EU ETS carbon price until 2030
(95 EUR/ton CO₂ eq.)⁴ in the TCO calculations (see Figure 3). While it makes a sizable
difference to the AHB's TCO, gray hydrogen remains less expensive than green

hydrogen, even when utilizing the most optimistic green hydrogen price of 3 EUR/kg.
As the price difference between the two types of hydrogen will keep getting smaller
and carbon prices are expected to rise in the 2030's and 2040's, green hydrogen will
become more and more competitive.

Please find the changes to the manuscript below (lines 150-159):

Next to the influence of all boat specific factors on the TCO, it is important to also
consider the impact of the carbon price. This regulatory mechanism, such as the
European Emissions Trading System (EU ETS), requires users to pay for their carbon
emissions, thereby favoring cleaner fuels like green hydrogen. With an average
projected EU ETS carbon price of 95 €/ton CO₂ eq. until 2030, it becomes evident that
operating the AHB with gray hydrogen will incur higher average additional costs
(0.10 €/km) compared to using green hydrogen (0.04 €/km). While introducing carbon
pricing is a way to move towards a more sustainable economy, the TCO calculations
show that the AHB operated by gray hydrogen is still significantly cheaper, requiring
even higher carbon prices to reach a break-even point.

**Comment 1.4**

I could not follow the argumentation of why the “end-of-life” phase was ignored in the
LCA, when this is clearly defined in ISO14040. To me this is like selling a car, but
without any wheels or engine. Do you mean that you are going to let the AHBs float
down the river when you are done with them? Isn't the whole point of doing an LCA to
include all phases of the life-cycle? Why do we otherwise have an ISO standard on
this if parts of it are going to be ignored? Of course you can buy your own engine and
wheels for your car that you bought without them, but it seems like a very strange
concept, and to the non-trained eye may provide distorted results with respect to a
proper LCA. In my opinion the results should be revised by taking the end-of-life phase
into account, and the discussion should be updated.

**Response 1.4**

We have now included the end-of-life phase into the LCA, which also entailed updating
our approach for calculating the TCO. Both now include a recycling phase for the metal
waste generated by the AHB at its end-of-life. Because of the complexity and
unforeseeable developments in recycling processes over the next thirty years, we have
kept our approach relatively simple (end-of-life material is treated entirely as aluminum
scrap since this makes up more than half of the mass across all scenarios) and have
chosen to not include recycling credits, therefore taking a conservative approach. The
weight of recycled material used for both LCA and TCO analysis is based on a bottom-
up approach targeted at the AHB's components found in the CAPEX or GWP100_{fixed}
respectively. To accurately account for the differing lifespans of individual components
compared to that of the AHB, we modeled the requisite replacements (such as the
renewal of PEM fuel cells every five years) by multiplying the component's weight by
the frequency of required replacements throughout the boat's operational lifespan.

Incorporating the recycling phase into the analysis results in minimal changes to the
TCO and across all impact categories of the LCA in all scenarios. This outcome was
anticipated due to the relatively low volume of materials requiring recycling at the end-
of-life.

Below, you can find all changes to the manuscript because of the inclusion of the
recycling phase:

Lines 102-104: Recycling costs reflect the costs for the disposal of end-of-life material
which is assumed to be entirely aluminum scrap to reduce complexity.

Lines 117-118: The GWP100 for recycling is calculated by modelling scrap aluminum
incineration.

Lines 305-308: As shown in equation (1), the LCPK of the AHB is, in principle,
calculated by dividing the sum of the annualized capital expenditures (CAPEX_{Annualized}),
the annual operational expenditures (OPEX), and the annualized recycling costs
(Recycling_{Annualized}) by the amount of km travelled per year.

$$\text{LCPK} = \frac{\text{CAPEX}_{\text{Annualized}} (1 + \text{maintenance rate}) + \text{OPEX} + \text{Recycling}_{\text{Annualized}}}{\text{km per year}} \quad (1)$$

Lines 363-366: Finally, the annualized recycling costs are calculated utilizing equation
(13), where the specific costs for the recycling of aluminum per kg are multiplied by the
total weight of end-of-life boat scrap and then divided by the lifespan of the boat. To
reduce complexity, the scrap is assumed to be aluminum scrap only.

$$\text{Recycling}_{\text{Annualized}} = \frac{C_{\text{Al recycling, specific}} \cdot \text{Scrap weight}_{\text{Total}}}{\text{Lifespan}_{\text{Boat}}} \quad (13)$$

Lines 389-391: The goal of this LCA is to analyze the environmental impact of
manufacturing, operating, and finally disposing of the proposed theoretical AHB at its
end-of-life. The system boundary can therefore be defined as being cradle-to-grave.

Lines 426-430: The overall GWP100 result per km (equation (14)) is calculated by
summing the fixed, variable, and recycling global warming potentials and then dividing
this sum by the total kilometers travelled per year. Again, this study included a
maintenance parameter set to 0.05 to stay consistent with the TCO analysis.

$$\frac{\text{GWP100 result per km}}{=} = \frac{\text{GWP100}_{\text{fixed}} (1 + \text{maintenance}) + \text{GWP100}_{\text{variable}} + \text{GWP100}_{\text{Recycling}}}{\text{km per year}} \quad (14)$$

Lines 457-460: Finally, GWP100_{Recycling} is calculated by multiplying the unit GWP100
impact of aluminum recycling per kg by the AHB's end-of-life scrap weight and then
dividing it by the total lifespan of the boat. To stay consistent, all scrap is assumed to
be aluminum scrap.

$$\text{GWP100}_{\text{Recycling}} = \frac{\text{unit_GWP100}_{\text{Recycling}} \cdot \text{Scrap weight}_{\text{Total}}}{\text{Lifespan}_{\text{Boat}}} \quad (23)$$

**Comment 1.5**

As a potential added value for this manuscript, it would be possible to combine the
effects of costs and greenhouse gas emissions in a marginal abatement cost (cost for
CO2 reduction between various options). Can you calculate the marginal abatement

cost with respect to road transport, for the green and gray hydrogen options? This
could be an interesting value for the reader, and can put the implementation of AHBs
into context with other green concepts/technologies.

**Response 1.5**

In the revised manuscript, we have added a marginal abatement cost analysis of the
AHB powered by both green and gray hydrogen compared to a diesel truck. For the
diesel truck, the TCO data comes from the NREL report which we have used
throughout our analysis⁵. Since the NREL paper does not take CO₂ emissions into
account, we have chosen to utilize data on average CO₂ emissions of real-world diesel
trucks in the EU⁶. More specifically, we used the CO₂ emissions for long-haul tractor-
trailers of the subclass 5-LH because they most closely resemble the trucks modeled
in the NREL paper.

The results of the marginal abatement cost analysis for both green and gray hydrogen
as discussed in the manuscript are provided below (lines 180-190):

A marginal abatement cost analysis for the AHB in scenario 2, powered by either green
or gray hydrogen, compared to a reference diesel truck (TCO only) and considering
average CO₂ emissions of diesel trucks (type 5-LH) in the EU28 reveals the following:
The AHB powered by green hydrogen has a marginal abatement cost of -0.17 €/kg
CO₂ eq compared to the diesel truck. In contrast, the AHB powered by gray hydrogen
demonstrates a substantially more negative marginal abatement cost of 4.64 €/kg CO₂
eq. While the highly negative marginal abatement costs for gray hydrogen powered
AHBs appear economically and ecologically advantageous, they do not reflect reality.
This is because reducing the carbon footprint of gray hydrogen would effectively alter
its classification, leading to entirely different cost structures (see S3 in the
supplementary materials for more details).

**Comment 1.6**

A clear overview of the various scenarios should be provided. For example: “The
speeds considered [for each of the scenarios] are 5 km/h, 10 km/h, 15 km/h, and 20
214 km/h and describe the speed of the AHB as seen on the global positioning system
(GPS).”

**Response 1.6**

We have revised the manuscript accordingly.

You can find the updated sentence below (lines 124-126):

The speeds considered in the four scenarios are 5 km/h, 10 km/h, 15 km/h, and 20
220 km/h and describe the speed of the AHB as seen on the global positioning system
(GPS).

**Comment 1.7**

“This, and the fact that the literature values for a fuel cell electric truck are based on
green hydrogen are the reason why scenario 2 for green hydrogen is compared more
closely to all three reference trucks in the next section.” This sentence was not clear to
me, please elaborate to explain more clearly what was meant.

**Response 1.7**

We have updated this sentence to make it clearer.

You can find the new version below (lines 173-176):

Because scenario 2 has the lowest GWP100 value compared to the other scenarios,
an adequate TCO, and the literature values for the reference fuel cell electric truck are
based on green hydrogen, this scenario is compared to the three reference trucks in
the next section.

Thank you again for your helpful comments!

**Reviewer #2**

**Comment 2.1**

The transport sector is large and diverse (long-distance ships, planes, trucks, etc.), but
this manuscript mainly focuses on long-haul trucking and alternatives. This should also
be mentioned in the first paragraph of the manuscript (the amount of goods and the
emissions they cause when being transported strongly depends on the type of
transport).

**Response 2.1**

The first two paragraphs of our manuscript set the scene and are therefore intentionally
broader. However, your comment made us realize that we need to articulate more
precisely what part of the transport sector we are focusing on.

In the revised manuscript, we now explain (lines 43-47):

To contribute to the field of transport research, this study focuses on a potential
alternative for long-haul trucking: the emerging technology of autonomous-driving
hydrogen-powered boats (AHBs). These boats could compete with semi-trucks for the
transportation of forty-foot equivalent unit (FEU) containers (one of the most common
means for transporting goods).

**Comment 2.2**

Only the benefits of AHBs are mentioned. I believe there are several downsides, such
as, but not limited to, the unclarity of the legal framework. Please mention these
downsides as well.

**Response 2.2**

AHBs do indeed face several challenges. In the revised manuscript we now list the
downsides you mentioned.

Please find this below (lines 47-50):

While AHBs, like other clean technologies in the early stages of their evolution, face
challenges such as technological and legal uncertainty as well as a lack of (hydrogen
refueling) infrastructure, they have potential for reducing carbon emissions in short to
medium-distance freight transport.

**Comment 2.3**

In the introduction, it remains unclear whether you compare the AHBs to conventional
inland ships or long-haul trucking.

**Response 2.3**

Your comment made us realize that we needed to make clearer that we compare the
AHB to long-haul trucking.

In the revised manuscript, we now write (lines 70-73):

**Second, by benchmarking the AHB against reference semi-trucks powered by diesel
fuel, batteries, and hydrogen fuel cells, this study is the first to derive potential
applications of AHBs for the decarbonization of the road freight sector.**

**Comment 2.4**

Figure 1 is tilted; please undo this.

**Response 2.4**

The figure is obtained from Python, utilizing vectorial world data from Natural Earth⁷.
Because the figure reflects the actual geographical borders of European countries,
what looks like a “tilt” is a direct result of utilizing the vectorial data, which we can
unfortunately not change. We kindly ask for your understanding.

**Comment 2.5**

A straightforward method is missing in this manuscript: The methods are described but
in several different sections. The readability of the manuscript would be significantly
enhanced by having one single method section (after the introduction and before the
results).

**Response 2.5**

It is journal policy to position the detailed method section after the conclusion. Broadly
explaining our approach in the main part is therefore intended to give the reader a
basic understanding of our approach, while leaving details to the method section as
291 per the submission guidelines of Communications Engineering.

**Comment 2.6**

Operational expenditures are broken down into fuel costs and loading costs (p 4, line
88): Where is the cost of human involved? Even though the ship is autonomous, people
will likely still be involved (also for legal reasons). How is this, compared to the truck?

**Response 2.6**

As with other emerging autonomous driving solutions (such as self-driving taxis) there
may be a period where a human driver is still legally required to monitor operation, but
in the medium- to long-term fully automated operation is expected (e.g., self-driving
taxis in several US cities no longer require a human driver). Fully automated operation
of boats has already been demonstrated by several companies. In the revised
manuscript, we now state that we assume fully autonomous operation of the boat.

Moreover, reflecting on your comment, we realized that we need to be clearer about
staff costs in general. We therefore clarified the loading parameter which not only
includes the action of loading and unloading but is also supposed to represent any
other costs related to human interaction to a certain extent. Since this was not
adequately addressed before, we have adjusted the description of $C_{Loading}$ in both the
main section and methods section of the manuscript. Legal costs are deliberately left
out at this point in time as these would be entirely speculative. Nonetheless, we believe
that the sensitivity analysis can shed some light on changes to the TCO when $C_{Loading}$
(which could at some point in the future also include legal costs by adjusting the
parameter) is varied. We have also added a sentence on future research opportunities
regarding the legal framework and costs to the conclusion section.

Here are the changes:

Lines 98-104: Loading costs are defined as a fixed monetary value and describe the
costs of loading and unloading a container from the AHB per trip. They, therefore,
capture costs for personnel involved and the use of port facilities. This study assumes
fully autonomous operation of the boat, which is why there are no staff costs for
navigating the boat.

Lines 285-286: Finally, future research could consider new legal frameworks and costs
needed to officially register AHBs.

Lines 360-362: The total loading costs (estimation of staff costs and port fees) per year
are determined by multiplying the fixed loading costs per trip with the ratio of km
travelled per year and average distance per trip (see equation (12)).

Regarding your second question about the truck: We compare our AHB data to a
previously published cost model of different types of future class 8 long-haul trucks
(diesel, battery-electric and hydrogen fuel cell) that are not modelled to be autonomous
and therefore include costs for human involvement in the form of driver wages⁵.

**Comment 2.7**

Line 93: better = lower?

**Response 2.7**

Correct. We have adjusted the manuscript accordingly (lines 97-98):

The higher the ratio of renewable to non-renewable energy, the lower the carbon
footprint of the green hydrogen.

**Comment 2.8**

Types of hydrogen: why only gray and green? What about the other colours (mainly
blue)?

**Response 2.8**

As this is something that reviewer #1 also noted, we have chosen to include blue
hydrogen in our analysis as well (see Response 1.2).

Below, you can find the specific changes to the manuscript as discussed in this section
(lines 191-204):

Besides green and gray hydrogen, blue hydrogen is another option. Like gray
hydrogen, blue hydrogen is produced through steam reforming of natural gas, but it
additionally incorporates carbon capture with efficiencies ranging from 50% to 95%,
thereby reducing greenhouse gas emissions. With an expected future price of around
2 €/kg, the TCO for AHBs using blue hydrogen could match that of gray hydrogen if
the carbon price is excluded. However, when the carbon price is included, blue
hydrogen offers superior performance due to its reduced emissions. Due to the
emission reduction, the AHB's GWP100 per km also improves, ranging from 0.64 kg
CO₂ eq/km at 50% carbon capture efficiency to 0.22 kg CO₂ eq/km at 95% efficiency.
These findings suggest that high carbon capture efficiencies may rival or exceed the
outcomes with green hydrogen. Despite the apparent advantages of blue hydrogen, it
should be noted that these calculations assume no additional emissions from the
auxiliary systems required for carbon capture. Including these in a comprehensive LCA
would likely result in less favorable outcomes.

**Comment 2.9**

Line 101 (life cycle assessment): how do you compare CO₂eq/GWP100 to money?

**Response 2.9**

Good point. We have adjusted this part and now explain (lines 109-110):

To compare the LCA and TCO analyses, the results of the LCA are normalized on a
362 per-kilometer-per-year basis, analogous to the TCO analysis.

**Comment 2.10**

I am missing a short method section where you summarize what exactly you are doing.
This may be a table, or a separate section, but it would be nice to immediately
understand what is in the model and how everything is calculated. Can be quite
compact I believe.

**Response 2.10**

Please understand that we must follow the journal's author guidelines, which is why
the method section is placed at the end of the manuscript before the references. We
believe that this section, together with the appendices, provide a comprehensive
overview of our approach.

**Comment 2.11**

Why do you consider the speeds (5-20km/h), and what are current speeds of e.g.
inland ships?

**Answer 2.11**

Current speeds of inland waterway ships are around 10-15 km/h on average⁸⁻¹⁰. To
provide the reader with a better understanding of how influential the speed is on the
costs and CO₂ emissions (also all other impact categories), we decided to add two
more extreme scenarios, one being 5 km/h (much slower than usual) and the other

being 20 km/h (faster than usual), to the two scenarios that are more realistic (10 km/h
and 15 km/h).

To clarify this in the manuscript, we have added a sentence as shown below (Lines
128-130):

**These speeds were chosen as they closely correspond to the average velocities (10**
**to 15 km/h) of common inland waterway vessels.**

**Comment 2.12**

Line 367: Why is decreasing the maintenance parameter deemed to be more
adequate?

**Answer 2.12**

Thanks for spotting this! We have now adjusted the parameter to 0.05 in the model
and text.

Below, you can find the revised sentence (line 429-430):

**Again, this study included a maintenance parameter set to 0.05 to stay consistent with**
**the TCO analysis.**

Thank you again for your helpful comments!

**References**

- 1. Frieden, F. & Leker, J. Future costs of hydrogen: a quantitative review.
*Sustainable Energy Fuels*; 10.1039/D4SE00137K (2024).
- 2. Gulli, C., Heid, B., Noffsinger, J., Waardenburg, M. & Wilthaner, M. Global
Energy Perspective 2023: Hydrogen outlook. *McKinsey & Company* (2024).
- 3. Ueckerdt, F. *et al.* On the cost competitiveness of blue and green hydrogen.
*Joule* **8**, 104–128; 10.1016/j.joule.2023.12.004 (2024).
- 4. BloombergNEF. EU ETS Market Outlook 1H 2024: Prices Valley Before Rally.
Available at [https://about.bnef.com/blog/eu-ets-market-outlook-1h-2024-prices-](https://about.bnef.com/blog/eu-ets-market-outlook-1h-2024-prices-valley-before-rally/)
[valley-before-rally/](https://about.bnef.com/blog/eu-ets-market-outlook-1h-2024-prices-valley-before-rally/) (2024).
- 5. Hunter, C. *et al.* Spatial and Temporal Analysis of the Total Cost of Ownership
for Class 8 Tractors and Class 4 Parcel Delivery Trucks, 2021.
- 6. Ragon, P.-L. & Rodríguez, F. CO2 emissions from trucks in the EU: An analysis
of the heavy-duty CO2 standards baseline data - International Council on Clean
Transportation, 25.01.2022.
- 7. Natural Earth. Downloads - Free vector and raster map data at 1:10m, 1:50m,
and 1:110m scales. Available at <https://www.naturalearthdata.com/downloads/>
(2024).
- 8. Karttunen, K., Väätäinen, K., Asikainen, A. & Ranta, T. The operational efficiency
of waterway transport of forest chips on Finland's Lake Saimaa. *Silva Fenn.* **46**,
395–413; 10.14214/sf.49 (2012).

- 9. ReWWay. Are inland vessels too slow to be competitive? Available at
<https://www.rewway.at/en/are-inland-vessels-too-slow-be-competitive/> (2024).
- 10. Union Pacific. Pros & Cons of Water Transport: Ship Speed, Shipment Visibility,
More. Available at [https://www.up.com/customers/track-record/tr082719-water-](https://www.up.com/customers/track-record/tr082719-water-pros-cons.htm)
[pros-cons.htm](https://www.up.com/customers/track-record/tr082719-water-pros-cons.htm) (2024).